# Biosolids-Derived Biochar Improves Biomethane Production in the Anaerobic Digestion of Chicken Manure

Soulayma Hassan [1,2], Tien Ngo [1,2], Leadin S. Khudur [1,2], Christian Krohn [1,2], Charles Chinyere Dike [1,2], Ibrahim Gbolahan Hakeem [2,3], Kalpit Shah [2,3], Aravind Surapaneni [2,4] and Andrew S. Ball [1,2,*]

1. School of Science, RMIT University, Melbourne, VIC 3083, Australia; s3885382@student.rmit.edu.au (S.H.); s3501132@student.rmit.edu.au (T.N.); leadin.khudur@rmit.edu.au (L.S.K.); christian.krohn@rmit.edu.au (C.K.); s3815496@student.rmit.edu.au (C.C.D.)
2. ARC Training Centre for the Transformation of Australia's Biosolids Resource, RMIT University, Bundoora, VIC 3083, Australia; ibrahim.hakeem@rmit.edu.au (I.G.H.); kalpit.shah@rmit.edu.au (K.S.); aravind.surapaneni@sew.com.au (A.S.)
3. School of Engineering, RMIT University, Melbourne, VIC 3000, Australia
4. South East Water, Frankston, VIC 3199, Australia
* Correspondence: andy.ball@rmit.edu.au

**Abstract:** Anaerobic digestion has attracted great interest for use in the management of organic wastes and the production of biomethane. However, this process is facing challenges, such as a high concentration of ammonia nitrogen, which affects the methanogenesis process and, thus, the production of methane. This study investigates the use of biosolid-derived biochar for mitigating ammonia stress and improving methane production during the anaerobic digestion of chicken manure, using both pristine biochar and biochar modified with a potassium hydroxide (KOH) solution. Batch mesophilic anaerobic digestion (37 °C) was carried out over 18 days. When compared to chicken-manure-only controls, a significant increase in methane formation was observed in the digesters amended with biochar and KOH-modified biochar, producing 220 L kg$^{-1}$ volatile solids (VSs) and 262 L kg$^{-1}$ VSs of methane, respectively, compared to 139 L kg$^{-1}$ VSs from the control digesters. The use of biochar and KOH-modified biochar resulted in a significant reduction of 8 days in the lag phase. The total ammonia nitrogen (TAN) concentration was reduced in the digesters with biochar and KOH-modified biochar by 25% and 35.5%, respectively. The quantitative polymerase chain reaction (QPCR) data revealed that the number of 16S rRNA gene copies was around 50,000 and 41,000 times higher in the biochar and KOH-modified biochar digesters, respectively, compared to the control digesters on day 18. The taxonomic profiles indicated that the BC and KOH-BC digesters contained a mixture of methanogenic pathways, including acetoclastic (*Methanosaetaceae*), hydrogenotrophic (*Methanosarcinaceae*), and methylation (*Methanofastidiosaceae*). This mix of pathways suggests a more robust archaeal community and, hence, more efficient methanogenesis. The results show that the addition of biosolids biochar enhances anaerobic digestion, mitigates ammonia stress to methanogens, and significantly increases biogas production.

**Keywords:** ammonia stress; adsorption; biogas; methanogenesis; microbial growth; pyrochar; waste materials

## 1. Introduction

Chicken manure (CM), a major by-product of the poultry industry, consists of a mixture of chicken faeces and bedding materials (generally sawdust, rice husks, and wheat straw) [1]. With the increase in the poultry breeding industry, large amounts of CM biomass are produced annually, with an estimated global production of 21,000 Mt [2]. This waste product can be used in farmlands as an organic fertiliser. However, a pre-treatment stage is required prior to soil application due to the negative impacts on the environment (greenhouse gas emissions, eutrophication, and odour arising from the decomposition

of CM) associated with the direct application of CM to the soil [3]. Alternatively, due to its high total solids content and high biodegradability, CM can be used as a feedstock in anaerobic digestion (AD) to produce biogas [2–4]. However, among all livestock manures, the AD of CM is considered to be the most challenging as it has the highest nitrogen (N) content [5]. The high percentage of N results in the build-up of $NH_3$, which leads to the inhibition of AD and, consequently, the inhibition of biogas production. The hydrolysis of uric acid and undigested proteins in CM is considered to be the main source of N, resulting in the production of amino acids [1]. During acidogenesis, the biodegradation of these amino acids releases significant amounts of N, which results in a build-up and ultimately results in the accumulation of $NH_3$ [6].

Free ammonia nitrogen (FAN/$NH_3$) and ammonium ions ($NH_4^+$) are the two forms of total ammonia nitrogen (TAN) present in aqueous environments [7]. Together, they are the main inhibitors of biomethane production during the AD of CM [5,7]. It has been reported that at an $NH_3$ concentration of 1500–2500 mg $L^{-1}$, inhibition can occur, resulting in a reduction in methane production [7]. Ammonia is considered toxic due to its physiochemical properties, such as its high solubility in lipids and its uncharged nature, which allows its diffusion across bacterial cell membranes [8]. Several methods are currently applied to mitigate $NH_3$ inhibition during the AD of CM. These methods include air stripping, which involves biogas recirculation to strip ammonia, and its simultaneous recovery using sulphate- or phosphate-containing receivers [9]. Bentonite can also be used to remove ammonia due to its significant swelling and adsorption capacity [10]. Moreover, trace element supplementation, such as $Fe^{2+}$ and $Ni^{2+}$, has been used to mitigate ammonia inhibition by enhancing the microbial communities and stimulating methanogenesis even when the ammonia concentration is high in the digesters [11]. Recently, the use of biochar as an additive in AD systems has attracted interest due to its benefits to the AD process. Biochar is a pyrolytic material rich in carbon derived from organic biomass [12]. This pyrolyzed carbon material has proven to be an excellent additive for several applications because of its widespread availability, low cost of production, and environmental friendliness [13]. Depending on its properties, the affinity of biochar towards organic and inorganic contaminants varies [14]. Several structural and physiochemical properties enable biochar to mitigate $NH_3$ inhibition, including richness in surface functional groups, cation exchange capacity (CEC), and high adsorption and porosity [3,15]. Biochar has surface functional groups, such as carboxyl (COOH) and hydroxyl (OH) [16]; due to electrostatic attraction, these groups form complexes with $NH_4^+$, which promotes the uptake of this cation [17]. Furthermore, the availability of metal elements in biochar enables the exchange of $NH_4^+$ with another cation, which results in surface adsorption. Once $NH_4^+$ is adsorbed to the surface, it is no longer bioavailable. Moreover, the porosity of biochar allows the colonisation and proliferation of microbial communities, particularly the methanogens within the pores. These pores offer shelter from $NH_3$ stress during the AD process, which increases the abundance of microbial communities in AD systems [18]. The additional growth of micro-organisms enhances the degradation of $NH_3$-N in the AD system. For example, Wang, Shi [19], He, and Han [20] reported that the addition of rice straw biochar could increase the relative abundance of some bacteria and archaea, including *Firmicutes*, *Bacteroidetes, Tenericutes*, *Synergistetes*, *Proteobacteria*, and *Sedimentibacter*, which enhanced the degradation of organic acids and ammonia–nitrogen in the anaerobic digestion system, resulting in an increase in biomethane production. All these factors have recently been highlighted by Cai, Zhu [21] to represent an integrated approach.

Biosolids are by-products of the wastewater treatment process, and their transformation to biochar and subsequent use in the AD of CM would be of environmental benefit and may provide new management pathways to the wastewater industry. The amount of biosolids produced annually is high and is projected to increase in the future due to population growth and urbanisation [22]. In 2021, Australia produced 349,000 tonnes of dry biosolids, which is 16% higher than the biosolids produced in 2010 (300,000 tonnes) [23]. The presence of contaminants in biosolids, such as per- and poly-fluoroalkyl (PFAs) sub-

stances and heavy metals, threatens its land application in agriculture, which is the largest end use of biosolids [24]. The pyrolysis of biosolids would be a solution, as it produces a useful biochar with reduced levels of PFAS and other contaminants [25]. Although the concentration of heavy metals is high in biosolids biochar, the bioavailability of most heavy metals from biosolids biochar is suggested to be very low [25,26]. An appropriate pyrolysis temperature has been shown to prevent the subsequent leaching of heavy metals, reducing their bioavailability [27]. In addition, biosolids-derived biochar contains micronutrients, such as Co, Ni, Mo, and Fe, which play important roles in enhancing microbial growth [28]. Biosolids biochar possesses properties that are suited to their application in agriculture and soil remediation; Stylianou, Christou [29] reported that biosolids biochar showed good stability, nutrient richness, and high calorific value. However, to date, to the best of the authors' knowledge, no study has used biosolids biochar to mitigate $NH_3$ stress in the AD of chicken manure nor assess its effects on biogas production. Different biomass biochar, such as that from wood, rice husk, fruitwood, and corn stover, have been previously shown to enhance the anaerobic digestion of CM [30–33]. However, since the parent material from which biochar was derived has a dominant role in determining its properties [34], there is a need to investigate biosolid-derived biochar for its impact on the AD of CM. The use of biochar in AD systems would not only create a potential market for biosolids to be used as additives but also help to reduce the amount of organic waste generated each year by providing a way to reutilise two waste streams (biosolids and chicken manure), which adheres to the principles of a circular economy.

The properties of biochar may influence its efficacy in anaerobic digestion and preventing $NH_3$ inhibition [21]. The modification of biochar also provides an opportunity to improve the properties of biochar for enhanced AD. Several studies have demonstrated that modification of biochar via acid and/or alkali increases the porosity and surface adsorption of biochar. For example, Vu, Trinh [35] reported that the number of surface carboxyl and sodium-containing functional groups of corncob-derived biochar increased following the modification of biochar with $HNO_3$ (6 M) and NaOH (0.3 M). This resulted in an increase in the surface adsorption capacity of the biochar, leading to a reduction in the ammonia concentration within the AD. In another study, the modification of corn stalk and rice hull biochar with 0.2 mol $L^{-1}$ $H_2SO_4$ led to a 1.57-fold increase in the adsorption capacity of the biochar [3]. This modification enhanced the electrostatic attraction between $NH_4^+$ and the surface functional groups of the treated biochar, resulting in an increase in $CH_4$ production.

The overall aim of the present study is to examine the impact of biosolids-derived biochar and KOH-modified-biosolids-derived biochar on the anaerobic digestion of CM. The specific objectives were to assess (i) the effect of unmodified and modified biosolids-derived biochar on biomethane production; (ii) the efficiency of biosolids biochar in reducing the level of total ammonia nitrogen (TAN) in digesters; (iii) the effect of biochar on the microbial (bacterial and archaeal) community structure and population; (iv) the impacts of biochar modification on the anaerobic digestion process.

## 2. Materials and Methods

### 2.1. Feedstock Collection and Preparation

CM was obtained from Bellarine Worms in Point Lonsdale, Victoria, Australia. Prior to use, CM was sieved to a particle size of <0.5 mm. Waste-activated sludge (S) was collected from the Mount Martha municipal wastewater recycling plant, South East Water Corporation, Melbourne, Australia, and was used as received. Biosolids biochar at <0.5 mm was used as an additive in this study. The biochar was obtained from the pyrolysis of digested and dewatered biosolids at 600 °C in a PYROCO pilot-scale pyrolysis plant in Melbourne, Victoria [36]. The production parameters and the physicochemical characteristics of the biochar used in this study were well-characterised in two prior studies [37,38].

## 2.2. Chicken Manure and Sludge Characterisation

The physicochemical properties of CM, S, and their mixture were studied in triplicate before the start of the experiment. Electrical conductivity (EC) (LAQUAtwin–EC–11, HORIBA Scientific, Kyoto, Japan), salinity (LAQUAtwin-Salt-11, HORIBA Scientific, Kyoto, Japan), pH HANNA Instruments edge$^{pH}$ (Keysborough, Victoria, Australia), and soluble chemical oxygen demand (COD) HACH Method 8000 (Loveland, CO, USA) were determined using a ratio of 1:10 (*w/v*) dry sample weight to Milli-Q deionised water. To determine the total ammonia nitrogen (TAN), a salicylate method was applied using the CM slurry and diluted sludge (1:100 dry weight to Milli-Q water), and a HACH DR 900 colourimeter (Loveland, CO, USA) was used to determine the TAN. Soluble chemical oxygen demand was determined according to Kassongo, Shahsavari [39], with analysis performed using a HACH DR 900 colourimeter. Total solids (TSs) were measured by placing 1 g of CM and 5 g of S in an oven at 105 °C for 24 h. Volatile solids (VSs) were determined by placing the dried samples in a furnace at 550 °C for 2 h; VSs were calculated as the difference between the initial and final weights of the samples. Table 1 summarises the physiochemical characteristics of CM, S, and their mixture CM+S (4.5:1 VSs ratio) used during the AD process.

**Table 1.** The physiochemical characteristics of chicken manure (CM), sludge (S), and chicken manure plus sludge (CM+S) at day 0, with their units.

| Characteristics | Unit | Chicken Manure | Sludge | Chicken Manure + Sludge |
|---|---|---|---|---|
| pH | – | 8.2 ± 0.0 | 7.5 ± 0.1 | 7.5 ± 0.1 |
| Salinity | % | 27.0 ± 0.0 | 4.5 ± 0.7 | 12.3 ± 0.6 |
| Electrical Conductivity | mS cm$^{-1}$ | 54.5 ± 0.8 | 8.4 ± 0.4 | 24.7 ± 1.2 |
| Moisture Content | % | 19.3 ± 1.6 | 98.3 ± 0.1 | 85.4 ± 0.8 |
| Total Solids | % | 80.6 ± 1.5 | 2.2 ± 0.0 | 14.6 ± 0.8 |
| Volatile Solids | % | 54.0 ± 1.7 | 1.7 ± 0.1 | 10.6 ± 1.0 |
| Total Ammonia Nitrogen | mg L$^{-1}$ | 3850.0 ± 353.0 | 1200 ± 0.0 | 2722.3 ± 195.4 |
| Soluble Chemical Oxygen Demand | g L$^{-1}$ | 37.7 ± 3.3 | 1.5 ± 0.2 | 14.9 ± 0.8 |

The results are the means of three replicates with the ± standard deviation shown.

## 2.3. Biochar Treatment and Characterisation

Biochar (200 g) was treated with 500 mL of 1 M KOH solution [40], and the mixture was agitated on a magnetic stirrer for 90 min at maximum speed and room temperature. The slurry was separated, and the solid residue was neutralised by washing with 4.5 L of Milli-Q water. The neutralised biochar was recovered by filtration using a Millipore 0.22 μm membrane filter. The modified biochar was dried in an oven at 100 °C for 24 h.

Fourier-transformed infrared spectroscopy (FTIR, Spectrum 100, Perkin Elmer, Shelton, CT, USA) was used for the assessment of the surface chemical functional groups of unmodified and KOH-modified biochar over the wavebands of 4000–650 cm$^{-1}$. The surface area of the biochar was analysed using Brunauer-Emmett-Teller (BET) surface area analysis using a Micrometritics TriStar II instrument (Norcross, GA, USA), according to Hakeem, Halder [24]. Inductively coupled plasma mass spectrometry (ICP-MS-700 Series, Agilent Technologies, Santa Clara, CA, USA) was used to measure the concentration of the trace metals in the biochar following acid digestion, according to method 3050B [41]. The surface morphology of the biochar was analysed using scanning electron microscopy (SEM, FEI Quanta 200 (Hillsboro, OG, USA) at ×5000 and ×10,000 magnifications, a 5.0 spot size, and a voltage of 30.0 kV.

### 2.4. Experimental Setup and Gas Sampling

Three treatments were run in duplicate: biochar (BC), KOH-modified biochar (KOH-BC), and the control (no biochar) for 18 days using 0.5 L Schott bottles. A duration of 18 days was chosen according to the findings by Shapovalov, Zhadan [42], where an industrial batch reactor (Biocel) was reported to have a typical retention time of 15–21 days. Sludge was used as inoculum for all treatments. Sludge alone was run in duplicate to determine the background methane production. The CM/sludge ratio was 4.5:1 VSs, and the ratio of CM to biochar was 2:1. The ratio of CM/biochar was chosen based on previous studies to achieve a biochar dosage of 5% TSs [30,43]. The total solids (TSs) with the addition of biochar was 15%. All digesters were operated at 37 °C to maintain mesophilic conditions. High-purity nitrogen was initially used to flush the Schott bottles to create anaerobic conditions. The digesters were shaken at 37 °C for 72 h before setting up the anaerobic experiment to achieve a homogenous mixture and ensure full anaerobic conditions. Trapped biogas was collected via water displacement and extracted every 24 h using a 50 mL syringe. An MX6 iBrid Portable Multi Gas Monitor (Pittsburgh, PA, USA) was used to analyse $CH_4$. Figure S1 in the Supplementary Materials section shows the experimental design used for the anaerobic digestion of chicken manure.

### 2.5. Biomethane Determination

Daily biomethane production was calculated using the percentage composition of $CH_4$ (% $CH_4$) and the total biogas volume (tBiogas). Methane production was calculated using the equation below, and the values were presented in L $kg^{-1}$ VSs [30].

$$\text{Daily } CH_4 = [\text{tBiogas treatment} \times (\% \, CH_4)] - [\text{tBiogas sludge} \times (\% \, CH_4)] \qquad (1)$$

### 2.6. Post-Digestion Chemical Analysis

At the end of incubation, all analyses were conducted in duplicate (EC, salinity, pH, TAN, and CODs) using the previously described methods and the same dilution ratios.

### 2.7. Biochar Extraction

Biochar and modified biochar were extracted from the digestates on day 18. Digestates (10 mL) were mixed with 40 mL of Milli-Q water in 50 mL centrifuge tubes and centrifuged for 5 min at 9500 rpm and 25 °C. The supernatant was discarded. The procedure was repeated three times until the supernatant became clear. Milli-Q water was added to the pellet, and the mixture was filtered using Millipore 0.22 μm membrane filters. Filtered biochar was placed in 2 mL microcentrifuge tubes and centrifuged for 5 min at 9500 rpm and 25 °C. The supernatants were discarded, and the pellets were stored for SEM analysis.

### 2.8. DNA Extraction

Digestate samples (0.25 g) were used to extract DNA using the DNeasy PowerSoil Kit (Hilden, Nordrhein-Westfalen, Germany), following the supplier's Protocol. A NanoDrop Lite Spectrophotometer (Hilden, Nordrhein-Westfalen, Germany) at absorbance ratios of 260/280 nm was used to screen the quantity and quality of the extracted DNA.

### 2.9. Real-Time Quantitative Polymerase Chain Reaction (qPCR)

A QIAGEN Rotor-Gene was used to quantify the number of gene copies present in the DNA from the extracted samples at the beginning and end of the incubation (Days 0 and 18). The procedure used was that described by Shahsavari, Aburto-Medina [44]. The primer sets 341F (CCTACGGGNGGCWGCAG) and 518R (ATTACCGCGGCTGCTGG) were used for the qPCR amplification of the 16S rDNA. The obtained data from qPCR was log-transformed prior to analysis.

*2.10. 16S rRNA Amplicon Sequencing for Bacterial Phyla and Archaeal Family Analysis*

The V4 region of the 16S rRNA gene was sequenced using extracted DNA samples. The sequencing was performed with the use of the primer set 341F (CCTACGGGNG-GCWGCAG) and 806R (GGACTACNVGGGTWTCTAAT). The protocols followed during the sequencing preparation were from Krohn, Jin [45]. The DNA concentrations were quantified using a Qubit 4.5 fluorometer from Invitrogen (Lenexa, KS, USA).

*2.11. Data Analysis*

The average and standard deviation of the triplicates and duplicates were calculated to present the data. The data were manipulated and analysed using MS Excel 365 and MINITABsoftware, version 19, respectively. A one-way analysis of variance (ANOVA) test was used for all experimental data. Any significant differences between the datasets were determined at $p < 0.05$.

## 3. Results and Discussion

*3.1. Effect of Biochar on Biomethane Production*

From day 1 to day 18, the digesters that had been amended with BC and KOH-BC produced significantly higher volumes of $CH_4$ compared to the control digesters ($p < 0.05$) (Figure 1). On day 18, the BC and KOH-BC digesters produced 220 L kg$^{-1}$ VSs and 262 L kg$^{-1}$ VSs, respectively, compared to 139 L kg$^{-1}$ VSs from the control digesters, representing an increase of 58.2% and 88.5%, respectively. A recent review reported that higher microbial cell density around a biochar leads to the increased hydrolysis of organic matter, resulting in a higher methane yield [46]. Another study reported that biochar can influence the action of several enzymes, such as dextranase, protease, and lipase, resulting in an increase in the degradation of organic matter [47]. It is likely that BC and KOH-BC increased $CH_4$ formation by accelerating the degradation of organic matter in the digesters. Similar observations were also made by Ngo, Khudur [30], where the addition of wood biochar and acid-alkali biochar improved the degradation of CM in AD by 77.5% and 89.1%, respectively. Pan, Ma [31] also reported an increase of 69% in methane production following the addition of fruitwood biochar to the AD of CM. No significant difference between the cumulative $CH_4$ production in BC and KOH-BC digesters was observed during the experiment ($p > 0.05$).

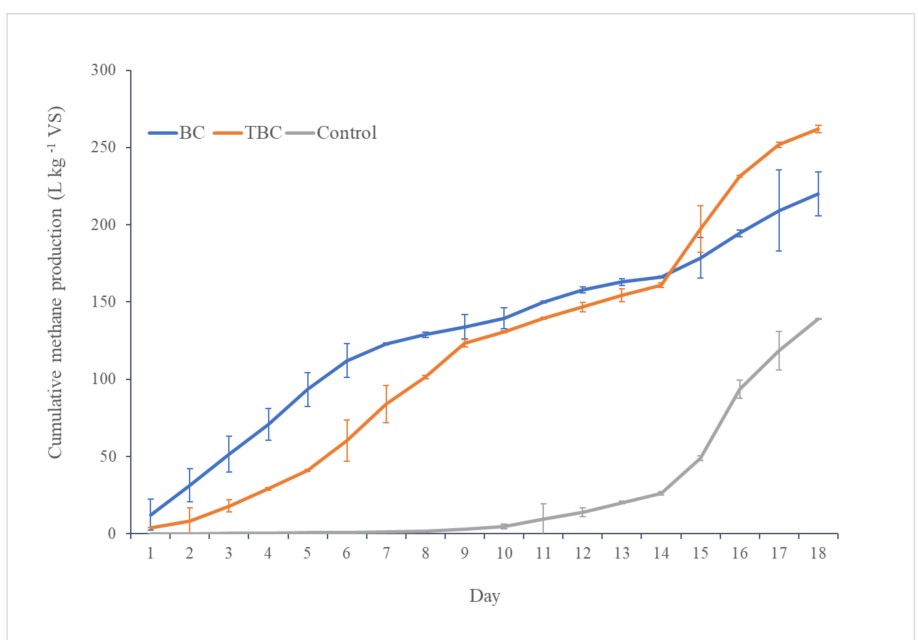

**Figure 1.** Cumulative methane production (L kg$^{-1}$ VSs) over 18 days for biosolids biochar (BC), modified biochar (KOH-BC), and no biochar (Control). The results are the means of two replicates, with error bars shown as $\pm$ standard deviation.

The addition of BC and KOH-BC also resulted in a significant reduction in the lag phase compared to the control ($p < 0.05$). The production of $CH_4$ started from day 1 in the BC and KOH-BC digesters, with 12.3 L kg$^{-1}$ VSs and 4.0 L kg$^{-1}$ VSs, respectively. However, the production of $CH_4$ in the control digesters did not start until day 8. Overall, the BC and KOH-BC digesters produced 129 L kg$^{-1}$ VSs and 101 L kg$^{-1}$ VSs, respectively, compared to 2 L kg$^{-1}$ VSs from the control digesters in the first 8 days. Luo, Lü [48] reported similar results, where the use of 0.5–1 mm biochar in digesters fed with 4, 6, and 8 g L$^{-1}$ glucose at 35 °C reduced the lag phase by 11.4%, 30.3%, and 21.6%, respectively, compared to the control digesters. Fagbohungbe, Herbert [49] also reported that increasing the biochar ratio shortens the lag phase prior to methanogenesis.

Overall, the use of BC and KOH-BC in the digesters enhanced the transformation of the macromolecules, which resulted in a rapid initial digestion. Traditionally, the duration of the lag phase is closely related to the recovery level of the microbial cells in digesters [49]. The addition of biochar stimulates cell growth, improves their enrichment and activity, and increases their metabolism during AD [50]. All these functions of biochar led to the acceleration of the hydrolysis, acidogenesis, and acetogenesis phases of the AD process. Pan, Ma [31] suggested that the use of fruitwood biochar would accelerate the transformation of macromolecules into dissolved substrates, which will be used to produce $CH_4$ at a faster rate than without biochar.

### 3.2. Effect of Biochar on Total Ammonia Nitrogen Reduction

Table 2 shows the TAN concentrations (in mg L$^{-1}$) at day 18 in all digesters (Control, BC, and KOH-BC). Changes in TAN (mg L$^{-1}$) and TAN Reduction (%) were calculated using the equations shown in the Supplementary Materials (Figure S3) and Methods section.

**Table 2.** Total ammonia nitrogen TAN (in mg L$^{-1}$) at day 18 in all digesters (Control, BC, and KOH-BC); change in TAN compared to day 0 mixture CM+S (in mg L$^{-1}$) and TAN reduction (%).

| Digesters | TAN at Day 18 (mg L$^{-1}$) | Change in TAN (mg L$^{-1}$) | TAN Reduction (%) |
|---|---|---|---|
| No Biochar (Control) | 2950 ± 195.4 | +227.7 | 0% |
| Biosolids Biochar (BC) | 2200 ± 0.0 | −522.3 | 25% |
| Alkali Biochar (KOH-BC) | 1900 ± 0.0 | −822.3 | 35.5% |

The results of TAN at day 18 are the means of two replicates with the standard deviation shown.

High concentrations of ammonia in digesters can lead to low digestion performance due to the imbalance of bacterial and archaeal micro-organisms, resulting in a low biogas yield [21]. It has been reported that biochar adsorbs ammonia nitrogen through several mechanisms, including chemical and physical adsorption [51]. Total ammonia nitrogen was measured at the start and end of the experiment (day 0 and day 18) to determine the change in the TAN concentration and the percentage of TAN reduction during the AD process (Table 2). Over the 18 days of the experiment, TAN reduced from 2722 mg L$^{-1}$ at day 0 to 2200 mg L$^{-1}$ and 1900 mg L$^{-1}$ in the digesters containing BC and KOH-BC, respectively. This significant decrease in TAN was higher in the KOH-BC digesters (30.2%) than in the BC digesters (19.2%) ($p < 0.05$). In contrast, an increase in TAN was observed in the control digesters (2950 mg L$^{-1}$). This value was significantly higher than TAN at day 0 by 8.4% ($p < 0.05$). The decrease in TAN concentrations in the digesters containing biochar indicates that both BC and KOH-BC were effective at adsorbing $NH_4^+$. Ammonia adsorption by BC and KOH-BC can be due to electrostatic attractions between $NH_4^+$ and the surface functional groups of biochar and/or a cation exchange of $NH_4^+$ with another positively charged element on the surface of the biochar [17]. Total ammonia reduction by biochar has been reported in several studies; Yu, Sun [33] demonstrated that the addition of rice-husk biochar reduced the TAN in digesters by 19.5%. In another study that aimed to compare the effects of different types of biochar on the AD of CM, Pan, Ma [31] reported that the use

of wheat straw and discarded fruitwood biochar reduced the TAN concentrations by 21% and 25%, respectively.

The modification of biochar is considered an effective approach to enhance its adsorption capacity by increasing the adsorption sites and surface area [51]. When compared to BC, KOH-BC addition resulted in greater TAN reduction (35.5%). This higher TAN removal efficiency can be explained by the physical and chemical changes of the biochar surfaces resulting from KOH treatment [52]. A similar result was reported by Carey, McNamara [40], where the biosolids biochar modified by KOH (1 M) adsorbed three times more TAN than the washed biochar.

### 3.3. Effect of Biochar on Digestates Properties

Several chemical tests were conducted on the digestates on day 18, including pH, EC, salinity, and CODs. The chemical characteristics of the digestates on day 18 were compared with the chemical characteristics of the mixture (CM+S) from day 0 in Table 1. Table 3 shows the physiochemical characteristics of the day 18 digestates and changes (%) compared to the day 0 mixture.

**Table 3.** Characteristics of day 18 digestates and changes (%) compared to day 0 mixture.

| Characteristics | Chicken Manure, Sludge, and No Biochar (Control) | | Chicken Manure, Sludge, and Biochar (BC) | | Chicken Manure, Sludge, and Alkali Biochar (KOH-BC) | |
|---|---|---|---|---|---|---|
| | Value | Change | Value | Change | Value | Change |
| pH | $8.1 \pm 0.0$ | N.A. | $8.0 \pm 0.0$ | N.A. | $8.0 \pm 0.0$ | N.A. |
| Electrical Conductivity ($mS\,cm^{-1}$) | $23.4 \pm 0.7$ | $-5.4\%$ | $16.2 \pm 1.1$ | $-34.6\%$ | $31.9 \pm 0.2$ | $+28.8\%$ |
| Salinity (%) | $11.5 \pm 0.7$ | $-6.7\%$ | $8.5 \pm 0.7$ | $-31.1\%$ | $12.5 \pm 5.0$ | $+1.4\%$ |
| Soluble chemical oxygen demand ($g\,L^{-1}$) | $20.9 \pm 3.7$ | $+40.8\%$ | $9.0 \pm 0.0$ | $-39.4\%$ | $8.9 \pm 0.8$ | $-40.4\%$ |

The results are the means of two replicates with the standard deviation shown. N.A.: not applicable.

Statistical tests confirmed no significant difference in pH in all digesters ($p > 0.05$); all digesters remained in the optimal range for ammonia adsorption (pH 7–8), where the adsorption of $NH_4^+$ is unaffected according to Khalil, Sergeevich [53]. The optimal pH range of 7–8 promotes surface adsorption by increasing the number of negatively charged functional groups on biochar, which results in an increase in electrostatic attraction between $NH_4^+$ and negatively charged groups. Yin, Liu [17] demonstrated that a pH of 8 was optimal to achieve the highest adsorption of $NH_4^+$.

Among the three digesters, EC was significantly reduced in the control and BC digesters, from 24.73 $mS\,cm^{-1}$ at day 0 to 23.4 $mS\,cm^{-1}$ and 16.2 $mS\,cm^{-1}$, a decrease of 5.4% and 34.6%, respectively ($p < 0.05$). In general, during AD, the degradation of macromolecules into smaller molecules results in a reduction in media conductance, which limits electron transfer to methanogenic micro-organisms [54]. Therefore, the maintenance of EC during AD would promote methanogenesis [39]. Electrical conductance increased significantly to 31.9 $mS\,cm^{-1}$ in the digesters containing KOH-BC, an increase of 28.8% ($p < 0.05$). A positive correlation was reported between EC and direct interspecies electron transfer (DIET) [55]. The increase in EC in the KOH-BC digesters suggests a possible increase in DIET, resulting in enhanced methanogenesis and, thus, higher $CH_4$ production.

Usually, EC can be used as a parameter to determine the soluble salt content in digesters. Controlling this parameter to a moderate level is important for the prevention of dehydration and the inhibition of methanogenic archaea in AD systems [10]. The statistical analyses confirmed that there was no significant difference in salinity in all digesters ($p > 0.05$). The salinity values obtained at day 18 for the control, BC, and KOH-BC digesters did not negatively affect $CH_4$ production. Ngo, Khudur [30] reported that despite

the increase in salinity in the digesters to 15%, this percentage did not appear to affect biogas production. Generally, a negative correlation is seen between salinity and $CH_4$ production; high salinity will result in the inhibition of acetate formation, as well as the cellular functioning of methanogens [56,57].

　　Soluble COD significantly increased in the control digesters from 14.9 g $L^{-1}$ at day 0 to 20.9 g $L^{-1}$ at day 18 ($p < 0.05$). However, CODs decreased in the BC and KOH-BC digesters to 9.0 g $L^{-1}$ and 8.9 g $L^{-1}$, respectively. A similar observation was reported by Aramrueang, Zhang [58], where the addition of biochar reduced COD by 76%. Overall, the physical and chemical properties of biochar were shown to enhance the degradation of organic matter. Moreover, biochar was demonstrated to improve the activity of hydrolase enzymes during the AD process [47]. Therefore, methanogens may utilise the readily organic alcohols, biodegradable sugars, and fatty volatile acids in the soluble fraction of digesters containing biochar, which contributes to a reduction in CODs and the lag phase, as shown in Figure 1. The percentage of CODs removal in the KOH-BC digesters (40.4%) appeared higher than the CODs removal in the BC digesters (39.4), though not at a statistically significant level ($p > 0.05$).

### 3.4. Changes in Biochar Characteristics following Modification

　　In order to further elucidate the mechanisms involved in the observed effect of biochar additions on biomethane production, analyses of the TAN concentrations and the microbial communities of the biochar and its alkali version were conducted (pH, EC, BET surface area analysis, ICP-MS analysis, and FTIR analysis). Table 4 summarises the key findings of these analyses.

**Table 4.** The characteristics of biosolids biochar (BC) and modified biochar (KOH-BC) with respective units.

| Characteristics | Unit | BC | KOH-BC |
|---|---|---|---|
| pH | NA | 9.7 | 9.3 |
| Electrical Conductivity | mS cm$^{-1}$ | 3.9 | 13.6 |
| BET surface area | m$^2$ g$^{-1}$ | 15.5 | 18.8 |
| ICP-MS metals concentration | | | |
| K | | 7662.9 | 11,629.5 |
| Na | | 5350.9 | 5173.4 |
| Mg | | 7894.7 | 8359.1 |
| Al | | 17,941.1 | 18,362.8 |
| Ca | | 90,346.4 | 95,502.0 |
| V | | 17.7 | 18.4 |
| Cr | | 26.3 | 28.0 |
| Mn | | 379.1 | 401.9 |
| Fe | mg kg$^{-1}$ | 22,902.6 | 24,109.0 |
| Co | | 4.1 | 4.4 |
| Ni | | 24.2 | 25.9 |
| Cu | | 996.5 | 1070.3 |
| Zn | | 1497.2 | 1547.8 |
| As | | 4.7 | 5.1 |
| Mo | | 5.0 | 5.6 |
| Cd | | 1.0 | 0.9 |
| Sb | | 0.6 | 0.7 |
| Ba | | 265.4 | 280.9 |
| Pb | | 29.6 | 30.6 |

　　The pH of KOH-BC decreased from 9.7 in BC to 9.3. This slight decrease in pH after treatment can be explained by an excessive wash of the modified biochar to remove any

residual KOH solutions, as described by Carey, McNamara [40]. The electrical conductivity increased from 3.9 mS cm$^{-1}$ in BC to 13.6 mS cm$^{-1}$ following treatment, leading to an increase of 28.8 % in EC in those digestates containing KOH-BC (Table 3).

The metal analysis using ICP-MS showed that biosolids biochar contains trace metals such as Fe, Ni, Co, Mo and Cd (Table 4). It has been reported that the addition of trace metals enhances the stability in AD systems at high ammonia concentrations [1,59]. Therefore, the presence of such trace metals in biosolids biochar added to the anaerobic system would provide stable digestion performance, resulting in an increase in CH$_4$ production. No differences in the biochar metal concentrations as a result of the KOH treatment were observed, with the exception of potassium (K); the concentration of K increased from 7662.8 mg kg$^{-1}$ in BC to 11,629.5 mg kg$^{-1}$ in KOH-BC, indicating that the treatment of biosolids biochar by KOH increased the concentration of K by 51.8% for KOH-BC. This increase enhances the cation exchange of the biochar with NH$_4^+$, resulting in an increase in surface adsorption, as observed in Table 2 [17]. Manure-derived biochar exhibits greater cation exchange capacity compared to other types of biomass biochar [60].

The composition of biochar's surface functional groups is a key determinant of its efficacy in terms of contaminant removal and nutrient retention [61]. Fourier-transform infrared spectroscopy was conducted to identify the main surface functional groups of BC and KOH-BC and to determine the variation in surface functional groups following alkali treatment. Figure S2 in the Supplementary Materials section shows the FTIR spectra of BC and KOH-BC. The FTIR spectra show six main functional groups present in both BC and KOH-BC (see Table S2 in the Supplementary Materials section). Overall, no changes in functional groups were detected between BC and KOH-BC, indicating that the alkali treatment did not affect the surface functional group composition of the biochar. In this study, the biosolids biochar was pyrolyzed at 600 °C, and its alkali version was not rich in carboxyl and hydroxyl groups when compared to other types of biochar, such as wood biochar [30] and wheat straw biochar [16], suggesting that the adsorption of NH$_4^+$ happened mainly via cation exchange.

### 3.5. Changes in Biochar Surface Morphology and Brunauer-Emmett-Teller Surface Area Analysis

SEM images of BC and KOH-BC were taken on day 0 at 5000× *g* magnification (Figure 2a,b). The surface morphology of BC is different to the surface morphology of KOH-BC; The SEM of BC shows a rough continuous surface with a granular structure, with the presence of small pores distributed over the surface (Figure 2a). However, the SEM image of KOH-BC shows a rough, discontinuous structure with many pores (Figure 2b). The pores distributed on the KOH-BC surface are much larger than the pores observed on the BC surface. The treatment of biochar with KOH increased the number and size of the pores on the surface of the biochar's particles. The Brunauer-Emmett-Teller surface area analysis revealed that the surface area of KOH-BC (18.8 m$^2$ g$^{-1}$) was 21.4% greater than the surface area of BC (15.5 m$^2$ g$^{-1}$) (Table 4). This result aligns with other findings, where KOH treatment improved the BET surface area of biochars [62,63]. Chemical treatment has been used to increase the porosity of biochar; the use of KOH as a chemical activator has been shown to result in the formation of micropores [64]. A positive correlation is reported between surface area, the number and size of pores, and the adsorption capacity of biochar [63,64].

Biochar's porosity is also considered a key factor in the physical interaction between biochar particles and the microbial community in AD. The pores provide habitats for microbes to proliferate [61]. The increase in the number and the size of pores facilitates the attachment and colonisation of microbes on biochar [63]. Hence, the modified biochar offers better shelter for methanogens, which would increase their chances of survival in ammonia-stressed environments. Moreover, a positive correlation between the level of porosity and the rate of ammonia adsorption has been previously reported [7].

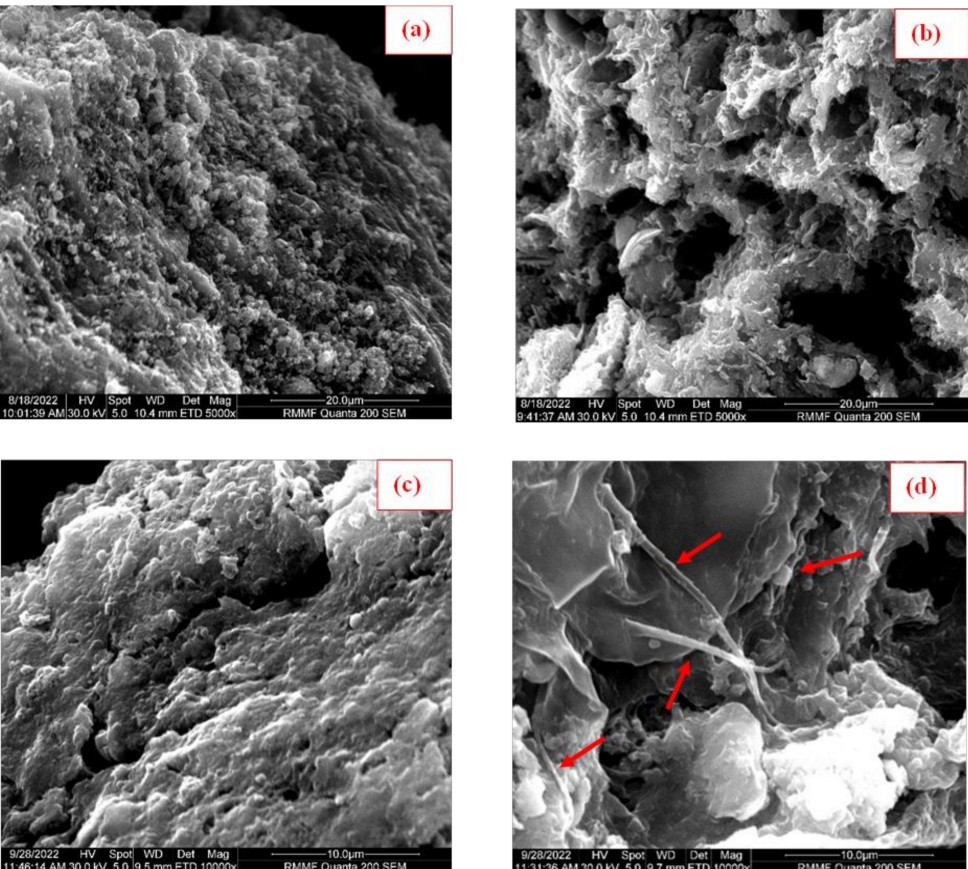

**Figure 2.** Scanning electron microscope (SEM) images of (**a**) biosolids biochar (BC) and (**b**) modified biochar (KOH-BC) taken on day 0 at 5000× *g* magnification; (**c**) biosolids biochar (BC) and (**d**) modified biochar (KOH-BC) taken on day 18 at 10,000× *g* magnification. The red arrows point out the microbial attachment.

In order to detect any microbial growth on BC and KOH-BC, SEM images of both biochars were taken on day 18 at 10,000× *g* magnification (Figure 2c,d). The surface morphology of BC at day 18 was similar to that at day 0 (Figure 2c). However, an SEM image of KOH-BC shows filamentous microbes, as well as round-shaped microbes, within the pores (Figure 2d). The increase in KOH-BC porosity would enhance the acclimation of methanogens during the anaerobic digestion process, with more sheltering and a lower level of ammonia stress.

*3.6. Effect of Biochar on Microbial Communities*

3.6.1. Quantitative Analysis of the Microbial Communities

Figure 3 shows the effect of adding BC and KOH-BC on microbial growth at the end of the incubation period (18 days). Overall, the addition of biochar promoted the growth of microbial communities in the BC and KOH-BC digesters; the qPCR analysis revealed that the number of 16S rRNA gene copies was 13,650 and 10,600 times greater in the BC and KOH-BC digesters, respectively, when compared to day 0 after 18 days of treatment, indicating a significant increase in the microbial biomass in the BC and KOH-BC digesters ($p < 0.05$). This increase was likely caused by two factors: (1) the reduction in TAN concentration by the biosolids biochar, which decreased the $NH_3$ stress on microbes, and (2) the sheltering offered by biochar to the microbial communities, which protected them from the ammonia toxicity in the digesters. A similar observation was reported by Lü, Liu [65], where the addition of pine biochar doubled the microbial biomass in digesters as a result of microbial acclimation. The increase in microbial population in the BC and KOH-BC digesters resulted in higher biogas production compared to the control digesters.

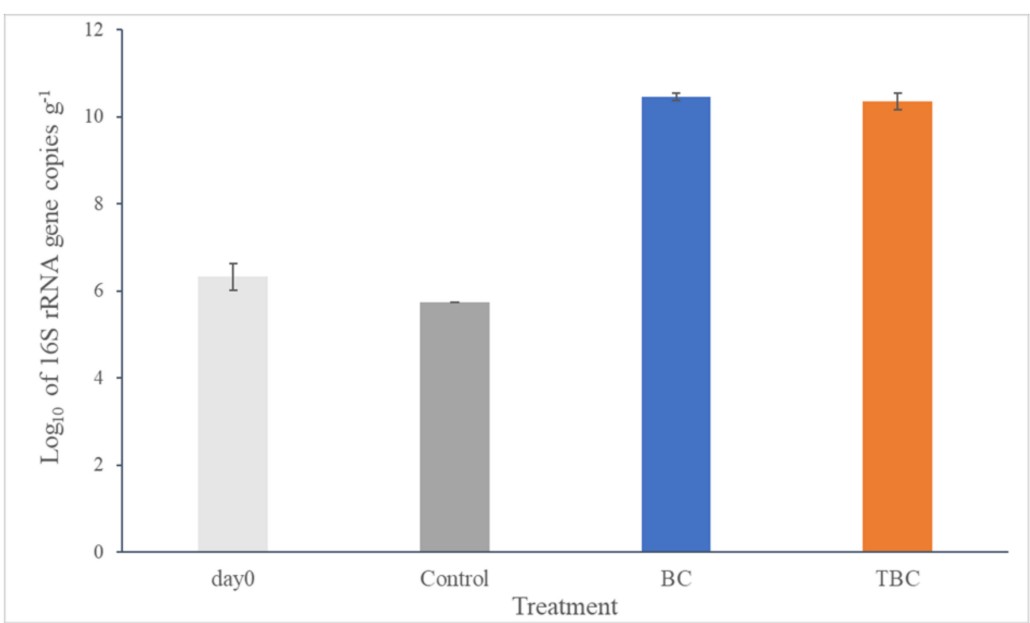

**Figure 3.** Changes in microbial communities ($\log_{10}$ of 16S rRNA gene copies $\text{g}^{-1}$) for the day 0 mixture and day 18 digestates: control (no biochar), biosolids biochar (BC), and modified biochar (KOH-BC). The results are the means of two replicates with error bars shown.

In contrast, the number of 16S rRNA gene copies on day 18 significantly decreased by 26% in the control digesters when compared to day 0 ($p < 0.05$), which was likely due to a high level of ammonia stress that negatively affected microbial growth. According to Sawayama, Tada [66], TAN concentration severely affects the growth of methanogens; high levels of ammonia in digesters cause ammonia toxicity, which alters the cellular functioning of methanogens and inhibits biogas synthesis. The number of 16S rRNA gene copies increased 50,000 and 41,000 times in the BC and KOH-BC digesters, respectively, when compared to the control digesters, confirming that the addition of biosolids biochar promoted the growth of microbial communities in the digesters, resulting in a higher biomethane yield.

3.6.2. Changes in Bacterial Phyla Induced by the Addition of Biochar

*Bacteroidota*, *Cloacimonadota*, and *Firmicutes* were the most abundant phyla in all digesters during the AD process, with a relative abundance of 19%, 34%, and 9%, respectively (Figure 4). According to the literature, these phyla are commonly dominant during the AD process [67]. *Bacteroidota* dominated all digesters by day 18, with a relative abundance (RA) increasing from 19% on day 0 to 52%, 40%, and 41% in the control, BC, and KOH-BC digesters by day 18, respectively. However, the relative abundance of *Bacteroidota* did not increase after 18 days. Similarly, the relative abundance of *Firmicutes* increased from 9% on day 0 to 23%, 22%, and 24% in the control, BC, and KOH-BC digesters by day 18, respectively. By day 18, the relative abundance of *Firmicutes* was 11%, which was a relatively small increase compared to the other digesters. It has been reported that *Bacteroidota* and *Firmicutes* play a crucial role in the AD process, especially in the hydrolysis and acidification stages [68,69]. Many studies demonstrated the importance of both phyla in the AD of CM [70,71]. It has been reported that *Bacteroidota* play a role in the decomposition of polysaccharides and, thus, the production of VFAs [72]. *Firmicutes* are known for their ability to decompose several substrates, such as polysaccharides, proteins, and lignocellulose, into VFAs [73,74]. In addition, *Firmicutes* are involved in microbial syntrophy during AD [75]. These bacteria degrade VFAs such as butyric acid and its analogues. The degradation results in the release of $H_2$, which can then be used by hydrogenotrophic methanogens to produce methane [75]. Ma, Chen [15] reported that *Bacteroidota* and *Firmicutes* had the main impact on the production of methane during AD. Therefore, the increase in relative abun-

dance of both phyla was desirable, as both phyla likely contributed to the decomposition of the CM and the production of methane.

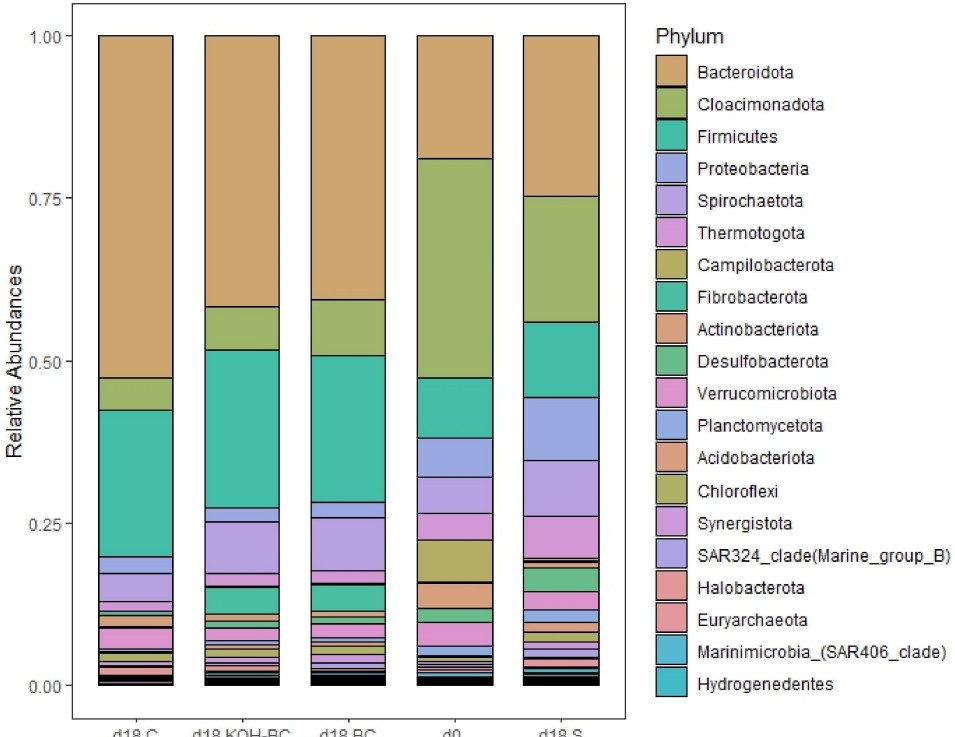

**Figure 4.** Relative abundance of the bacterial communities at the phylum level in the digestates on day 0 (d0) and on day 18: control (d18 C), biosolids biochar (d18 BC), modified biochar (d18 KOH-BC), and sludge (d18 S). The results are the mean of two replicates with no error bars shown.

The relative abundance of *Cloacimonadota* decreased from 34% on day 0 to 5%, 9%, and 7% in the C, BC, and KOH-BC digesters by day 18, respectively. The relative abundance of *Cloacimonadota* also decreased to 19% in the sludge digesters (d18S); however, this decrease was minimal compared to the other digesters. It has been reported that members of this phylum can degrade long-chain fatty acids [76]. Moreover, *Cloacimonadota* is involved in the hydrolysis of cellulose. Perman, Schnürer [68] reported that *Cloacimonadota* grows better in an environment with lower protein, fat, and sugar.

The addition of biosolids biochar did not impact the bacterial communities; similar communities were present in the control and biochar treatments. However, the addition of biochar may produce more pronounced effects on the methanogenic population, which will be discussed in the following section.

3.6.3. Changes in the Archaeal Population Induced by the Addition of Biochar

In order to better understand the changes in the archaeal population during the experiment, the relative abundance of the archaeal population was analysed. Figure 5 shows changes in the relative abundance of the archaeal communities at the family level for feedstock on day 0 and the control, BC, KOH-BC, and sludge digestates on day 18.

At the family level, the relative abundance of *Methanofastidiosaceae* decreased on day 0 from 36% to 3%, 12%, and 8% in the control, BC, and KOH-BC digesters, respectively, but increased to 67% in the sludge digesters by day 18 (Figure 5). It has been reported that *Methanofastidiosaceae* lack the proteins responsible for acetate and $CO_2$ metabolism and, hence, dominate low acetate environments [75]; this may explain their growth within the day 18 sludge digester. The relative abundance of *Methanosarcinaceae*, which are acetoclastic and hydrogenotrophic methanogens, was <1% at day 0 but grew over 18 days to reach 45%, 3%, and 21% in the C, BC, and KOH-BC digesters, respectively. However, the relative abun-

dance of *Methanosarcinaceae* remained <1% in the sludge digesters at day 18, likely due to the lack of acetate. *Methanosarcinaceae* were more abundant in the control rather than in the BC and KOH-BC digesters (Figure 5). It has been reported that *Methanosarcinaceae* spp. acclimatise well at high ammonia levels due to their coccoid shape and smaller surface area, which enable them to form clusters in high ammonia environments, reducing the free diffusion of ammonia [77]. *Methanosaetaceae* are strict acetoclastic methanogens and can be coccoid or filamentous. Due to their longer shape and inability to form clusters, *Methanosaetaceae* are sensitive to high ammonia concentrations. Their relative abundance decreased significantly from 27% on day 0 to 4% in the control digesters by day 18. *Methanosaetaceae* maintained their population in the digesters containing BC (RA:27%) and KOH-BC (RA:24%). This can be explained by the reduction in the TAN levels in the BC and KOH-BC digesters, as well as the microbial sheltering effect offered by biosolids biochar.

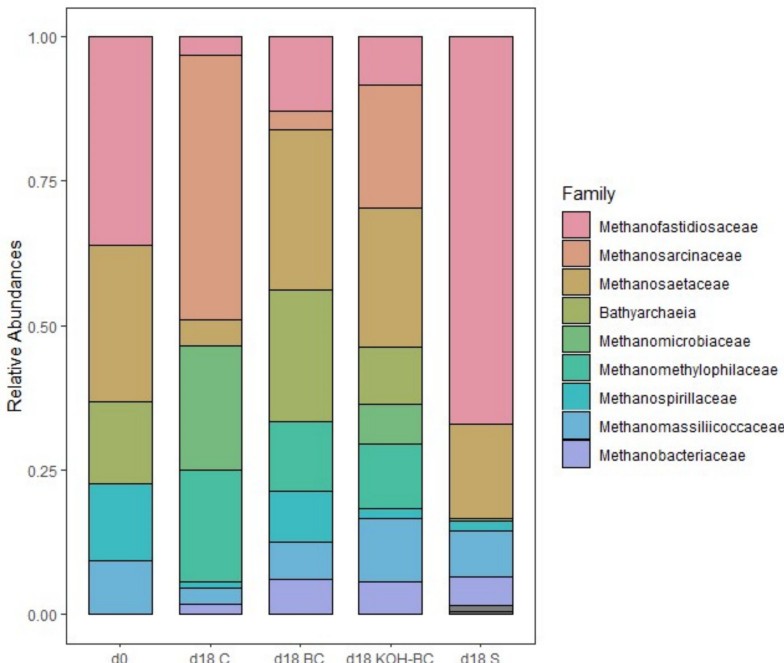

**Figure 5.** Relative abundance of the archaeal communities at the family level in the digesters on day 0 (d0) and the digestates on day 18: control (d18 C), biosolids biochar (d18 BC), modified biochar (d18 KOH-BC), and sludge (d18 S). The results are the mean of two replicates with no error bars shown.

The taxonomic profiles indicated that the BC and KOH-BC digesters contained a mixture of methanogenic pathways, including acetoclastic (*Methanosaetaceae*), hydrogenotrophic (*Methanosarcinaceae*), and methylation (*Methanofastidiosaceae*). This mix of pathways suggests a more robust archaeal community and, hence, more efficient methanogenesis. The production of methane in the digesters containing biosolids biochar was higher than the methane production in control digesters. A comparison between BC and KOH-BC indicated that KOH-BC might have favoured both *Methanosarcinaceae* and *Methanosaetaceae* populations, whereas BC might have only enriched the *Methanosaetaceae* populations. This finding agrees with the SEM images of KOH-BC, showing both round and filamentous-shaped microbes on KOH-BC. Both findings suggest that KOH-BC had a stronger microbial sheltering effect than BC.

## 4. Conclusions

This study demonstrated that biosolid biochar can produce similar outcomes in anaerobic digestion when compared to other biomass biochar. Biosolids biochar was effective in reducing the TAN concentrations in the digesters, protecting the microbial communities against ammonia stress, and promoting their proliferation. These integrated factors

resulted in an increase in $CH_4$ production and enhanced the degradation of organic matter in the digesters. Over the 18 days of anaerobic digestion, methane production was significantly higher in the digesters containing BC and KOH-BC (58.1% and 88.3%, respectively). The addition of biochar to the digesters resulted in a reduction in the lag phase and an acceleration of biogas production. The alkali treatment of biosolids biochar resulted in an increase in surface area and the number and size of the pores; this enhanced TAN removal by 35.5%. With a higher porosity, KOH-BC also provided shelter for microbial communities, protecting them from ammonia stress. Furthermore, alkali modification of biochar led to an increase in K, which would improve the cation exchange capacity of the modified biochar. However, alkali treatment did not affect the surface functional group composition of the biochar, indicating that the increase in the adsorption of $NH_4^+$ occurred mainly via cation exchange. The addition of biosolids biochar resulted in the creation of a mixture of methanogenesis pathways, including acetoclastic, hydrogenotrophic, and methylation, during digestion and, hence, more efficient methane production. Biosolids biochar is a viable, cost-efficient material that can be applied at low concentrations to enhance the anaerobic digestion of organic matter such as chicken manure.

Given that only one pre-treatment of biosolids biochar was investigated, future studies on the anaerobic digestion of chicken manure should investigate the use of other modified biosolids biochar, such as the acid-alkali version, and its effects on ammonia stress mitigation during the anaerobic digestion process. In addition, studies should be performed in continuous reactors, rather than small-scale batch reactors, to fully explore the long-term efficacy of ammonia adsorption and the microbial sheltering mechanisms of biosolids biochar and its alkali-modified version.

**Supplementary Materials:** The following supporting information can be downloaded at: https://www.mdpi.com/article/10.3390/resources12100123/s1. Figure S1: Experimental design of the anaerobic digestion of chicken manure, Figure S2: Fourier Transformed Infrared (FTIR) spectra of biosolids biochar (BC) and modified biochar (KOH-BC), Figure S3: Change in Tan (mg $L^{-1}$) and TAN Reduction (%) equations, Table S1: The mass and volume of materials added to digesters with different treatments, Table S2: The main surface functional groups of biosolids biochar (BC) and modified biochar (KOH-BC) identified from FTIR spectra.

**Author Contributions:** S.H.: Conceptualisation, Methodology, Visualisation, Formal analysis, Investigation, Data curation, Validation, Writing—original draft preparation, Writing—review and editing; T.N.: Conceptualisation, Methodology, Software, Formal analysis, Data curation, Validation, Writing—review and editing; L.S.K.: Methodology, Resources, Project administration, Validation, Writing—review and editing; C.K.: Methodology, Resources, Software, Project administration, Validation, Writing—review and editing; C.C.D.: Validation, Writing—review and editing; I.G.H.: Methodology, Resources, Validation, Writing—review and editing; K.S.: Resources, Validation, Writing—review and editing, Supervision; A.S.: Resources, Supervision; A.S.B.: Resources, Validation, Project administration, Writing—review and editing, Supervision. All authors have read and agreed to the published version of the manuscript.

**Funding:** This research received no external funding.

**Data Availability Statement:** The datasets used or analysed during the current study are available upon request.

**Acknowledgments:** The authors would like to acknowledge the RMIT Microscopy and Microanalysis Facility for the provision of the scanning electron microscope.

**Conflicts of Interest:** The authors declare no conflict of interest.

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
