# Peer review of "Biosolids-Derived Biochar Improves Biomethane Production in the Anaerobic Digestion of Chicken Manure"

_resources, doi:10.3390/resources12100123_

Round 1

Reviewer 1 Report

Apart from a few technical issues, the work is of good scientific and technical quality. Please, refer to the respective comments given in the pdf-file.

What bothers me is that I cannot find references 55 to 84 in the manuscript.

Since the manuscript was written by native English speakers, there are only marginal linguistic corrections to be made.

Reviewer 2 Report

The article is well written, but a few important questions need to be addressed or elucidated further as described below to make the article more useful from a practical implementation point of view of the proposed use of biochar in the anaerobic digestion of chicken manure.  The "Abstract" summarizes well the findings of the research effort. The "Introduction" describes succinctly the problem at hand and the various solutions that have been proposed during the past two decades.  The "Materials and Methods" while it describes comprehensibly the research effort, leaves out, in my view, or at least does not address explicitly a few critical elements to make biochar a viable means of addressing the anaerobic digestion of chicken manure.  These elements are as follows:  (a) The amount of energy required to produce biochar from dewatered sludge, which is quite wet or alternatively how much biochar can be produced from a given amount of dewatered sludge, assuming that the carbon in the sludge provides the required energy - this biochar production could be energetically quite significant; a corollary to this question is why you have chosen sludge biochar? You may wish to expand on that; (b) It appears (line 183) that a ratio of CM to Biochar of 2:1 has been used in the experiments - this is a relatively huge amount of biochar vis-a-vis the available chicken manure and would suggest that the use of biochar could not be a universal remedy - please also expand on the text lines 182 and 183, because it is not clear what amounts of CM and S were used for the control, BC and KOH-BC experiments - adding a table would be very beneficial to the reader to understand the exact composition details of what was actually tested as lines 182-183 leave much to be desired; (c) The addition of significant amounts of KOH to biochar increases further the cost of the process and it may not be worth the benefit in the reduction of TAN per Table 2 - please comment on that.  The "Results and discussion" section shows that the methane production started right away in the BC and the BC-KOH cases but had a lag time of 8 days in the control case (Figure 1).  You seem to claim that lag is in the delay of the degradation of macromolecules in the control compared to that in the presence of BC or BC-KOH.  Please explain how the presence of BC degrades/ dissolves the macromolecules in CM, etc.  Also, the increase in methane production in the control seems to take off after 14 days, but you stoped your experiment at day 18 to show the advantages of BC and BC-KOH tests.  Why did you stop at 18 days and what would have happened if you continued to, say, 30 days?  Lastly, while biochar can promote the digestion of CM, I am not convinced that the biochar has much to do with NH4+ or TAN, which is a key premise of your article.  I am suspecting the BC and BC-KOH act as stable surface matrices where the bacteria remain immobilized and can grow without being washed away thereby degrading the CM more effectively. This is also supported by your findings in Section 3.6 of the text. Please elaborate further.  The "Conclusions" section ascribes the increased methane production in the presence of BC and BC-KOH to the removal of TAN, which is in my view questionable as being the primary factor - see also my previous comments.  

General Comments:  (A) Improve the readability of all Figures - the captions along the axis are hardly legible; (B) I have been unable to match the numbering of references in the text in Section 2 and Section 3 with the corresponding numbers in the "References" section.  Several examples follow - please recheck: line 142 ref [1]; one 153: Kassongo, Shahsavari [2]; line 164: [3]; line 173 Hakeem, Halder [4]; line 214: Shahsavari, Aburto-Medina [7]; line 222: Krohn, jin [8]; line 238: Pan, Ma [10]; line 253: Luo,Lu [11]; line 255: Fagbohungbe, Herbert [12]; line 260: Pan, Ma [10]; line 287: Pan, Ma [10]; line 284: Carrey, McNamara [3]; line 306: Khahil, Sergeevich [16]; line 309: Yin, Liu [13]; line 358: Carrey, McNamara [3]; line 467: Ma, Chen [48]; line 476:  Pernan, Schnurer [38].  If I am correct, then all the reference numbers in the text must be checked against the references listed in the "Reference" section.  

Reviewer 3 Report

Comments can be found in the Reviewer's Comments file.

Reviewer 4 Report

 Reviewer Comments:

Graphical Abstract: Not Provided, suggestion to provide graphical abstract with 3D style so that it is more attractive.

Tables: If possible, try to include more meaningful tables.

Figures: If possible, try to use higher definition images.

Line Numbering: It is suggested that line numbers are added the side of each page, aligned with the main text.

General comment:

This paper focused on the impact of pristine and KOH-modified biosolids-derived biochar on the anaerobic digestion of chicken manure. Also, authors need to perform critical analysis and interpret all these studies and come up with a conclusion for each section. It’s good that you had this finding written but readers would preferable want to know what had you concluded from all these studies, instead of what the author of the literature studies had concluded. To conclude, this paper needs to revise it carefully before it can be considered in high impact journal. Hope below comments will be able to help to further improve the paper.

Specific comment:

Abstract:

-      Suggest modifying the title to be more attractive and related to the current trends.

-      Needs major revision prior to the amendment of the main content.

-      An abstract is often presented separately from the article, so it must be able to stand alone. Hence the problem statement, aim, novelty and results of the study, all should be included into the one paragraph of abstract.

-      Please try to merge all information into a paragraph with some attractive and new findings.

Keywords:

-       Kindly modify the keywords are not the same as the title, because keywords are a tool to help indexers and search engines find relevant papers. If database search engines can find your journal manuscript, readers will be able to find it too. This will increase the number of people reading your manuscript, and likely lead to more citations.

Introduction:

-      Introduction should be covered the gap of the research. However, it is not well covered in this section, it needs to be more specific instead of using only one sentence to cover that.

-      Also, please mention the important of this study to society as well as industry.

-      Page 1-2, Line 31-52; it is suggested that both paragraphs can be combined and simplified.

-      Page 2, Line 47; please do check if a subscript or superscript required for “NH3”.

-      Page 2, Line 53; please do use proper chemical formulas for ammonia, ammonium ions.

-      Page 2, Line 53-71; the elaboration on effect of NH3 and NH4+ is a bit too long, try to fit it into 3-4 sentences.

-      Page 2-3, Line 94-112; it is suggested both biochar and biosolids can be elaborated in a single shorter paragraph.

-      Introduction section is a bit lengthy, kindly simplify it.

-      Current introduction looks a bit too lengthy kindly revise the structure of the section.

-      Please revised the Introduction section based on the structure below that can make it more clearly:

1st paragraph: Problem statement

2nd paragraph: Current ongoing solution

3rd paragraph: Proposed solution in this work.

4th paragraph: Summarized the current research novelty and objective of this work.

-      Problem statement of your introduction is not strong, need to discuss more about it.

-      The objective and purpose of the study is not clear.

-      A good introduction should conclude the introduction by mentioning the specific objectives of the research.

-      Kindly refer to those additional materials to revise this section: (1) Bioenergy production from chicken manure: a review; (2) Biofilters and bioretention systems: the role of biochar in the blue-green city concept for stormwater management: Substitutions of petroleum-based surfactants; (3) Improving bioenergy production in anaerobic digestion systems utilising chicken manure via pyrolysed biochar additives: A review; (4) Green catalyst derived from zero-valent iron onto porous biochar for removal of Rhodamine B from aqueous solution in a Fenton-like process

-      The earlier paragraphs should lead logically to specific objectives of the study.

-      Note that this part of the Introduction gives specific details: for instance, the earlier part of the Introduction may mention the importance of this study whereas the concluding part will specify what methods of control were used and how they were evaluated.

Experimental Methods:

-      Authors need to specify the source of materials were used.

-      Also, authors are encouraged state the location of the equipment in this report.

-      Page 5, Line 195; please use proper caption (i.e. Eqn(1) or (1)) for equations.

-      Please elaborate more on the methodology as current description of method is not easy to understand and readers hard to repeat the analysis.

-      Please provide an additional figure to illustrate the process of the whole methodology.

Results and discussion sections

-      The overall structure needs to be improved.

-      If possible, it is advised that both theory and discussions, experimental results and discussions can be combined into single section to allow better elaboration of data based on each analysis conducted and results obtained.

-      Kindly improve on the discussion. What is the significance of the results of the work?

-      For section 3.1; it is mentioned that both BC and KOH-BC enhanced the transformation of macromolecules, but why is there a difference in improvement between both cases?

-      For section 3.3; how does the mentioned parameters impact the biomethane production through AD?

-      Current explanation on the result obtained is insufficient, more details and comprehensive elaboration are required.

-      There are some important information from this paper that authors are recommended to refer: (1) Effects of different types of biochar on the anaerobic digestion of chicken manure; (2) Role of biochar surface characteristics in the adsorption of aromatic compounds: Pore structure and functional groups; (3 Impact of biochar on anaerobic digestion: Meta-analysis and economic evaluation; (4) A critical review on biochar for enhancing biogas production from anaerobic digestion of food waste and sludge

-      It is quite doubt on some figures. Please recheck.

Conclusions

-      It is suggested that a challenged faced and future directions section can be added.

-      Current conclusions are a bit too lengthy, kindly revise.

-      It is suggested to include additional information or clarifications to the methods and results sections to evaluate the manuscript's novelty and its significance to the field.

-      Kindly improve to more concise with significant results.

References

-      Kindly revise reference format according to the author guideline.

-      It is suggested to cite references within 5 years of research to maintain the reliability of results obtained.

-      Please do use uniform line spacing throughout the manuscript.

-      There are references found to be outdated.

Extensive editing of English language required

Round 2

Reviewer 3 Report

Line 15: Please delete 'for the first time' from the sentence.

Reviewer 4 Report

The manuscript is corrected and revised according to the reviewer's comments. I am now satisfied with the new version, so I would like to recommend its publication.

Minor editing of English language required
